# Plan Better Amid Conservatism: Offline Multi-Agent Reinforcement Learning with Actor Rectification

## Abstract

The idea of conservatism has led to significant progress in offline reinforcement learning (RL) where an agent learns from pre-collected datasets. However, it is still an open question to resolve offline RL in the more practical multi-agent setting as many real-world scenarios involve interaction among multiple agents. Given the recent success of transferring online RL algorithms to the multi-agent setting, one may expect that offline RL algorithms will also transfer to multi-agent settings directly. Surprisingly, when conservatism-based algorithms are applied to the multi-agent setting, the performance degrades significantly with an increasing number of agents. Towards mitigating the degradation, we identify that a key issue that the landscape of the value function can be non-concave and policy gradient improvements are prone to local optima. Multiple agents exacerbate the problem since the suboptimal policy by any agent could lead to uncoordinated global failure. Following this intuition, we propose a simple yet effective method, Offline Multi-Agent RL with Actor Rectification (OMAR), to tackle this critical challenge via an effective combination of first-order policy gradient and zeroth-order optimization methods for the actor to better optimize the conservative value function. Despite the simplicity, OMAR significantly outperforms strong baselines with state-of-the-art performance in multi-agent continuous control benchmarks.

## 1 Introduction

Offline reinforcement learning (RL) has shown great potential in advancing the deployment of RL in real-world tasks where interaction with the environment is prohibitive, costly, or risky (Thomas, 2015). Since an agent has to learn from a given pre-collected dataset in offline RL, it becomes challenging for regular online RL algorithms such as DDPG (Lillicrap et al., 2016) and TD3 (Fujimoto et al., 2018) due to extrapolation error (Lee et al., 2021).

There has been recent progress in tackling the problem based on conservatism. Behavior regularization (Wu et al., 2019; Kumar et al., 2019), *e.g.*, TD3 with Behavior Cloning (TD3+BC) (Fujimoto & Gu, 2021), compels the learning policy to stay close to the manifold of the datasets. Yet, its performance highly depends on the quality of the dataset. Another line of research investigates incorporating conservatism into the value function by critic regularization (Nachum et al., 2019; Kostrikov et al., 2021), *e.g.*, Conservative Q-Learning (Kumar et al., 2020), which usually learns a conservative estimate of the value function to directly address the extrapolation error.

However, many practical scenarios involve multiple agents, *e.g.*, multi-robot control (Amato, 2018), autonomous driving (Pomerleau, 1989; Sadigh et al., 2016). Therefore, offline multi-agent reinforcement learning (MARL) (Yang et al., 2021; Jiang & Lu, 2021) is crucial for solving real-world tasks. Observing recent success of Independent PPO (de Witt et al., 2020) and Multi-Agent PPO (Yu et al., 2021), both of which are based on the PPO (Schulman et al., 2017) algorithm, we find that online RL algorithms can be transferred to multi-agent scenarios through either decentralized training or a centralized value function without bells and whistles. Hence, we naturally expect that offline RL algorithms would also transfer easily when applied to multi-agent tasks.

Surprisingly, we observe that the performance of the state-of-the-art conservatism-based CQL (Kumar et al., 2020) algorithm in offline RL degrades dramatically with an increasing number of agents

as shown in Figure 1(c) in our experiments. Towards mitigating the degradation, we identify a critical issue in CQL: solely regularizing the critic is insufficient for multiple agents to learn good policies for coordination in the offline setting. The primary cause is that first-order policy gradient methods are prone to local optima (Nachum et al., 2016; Ge et al., 2017; Safran & Shamir, 2017), saddle points (Vlatakis-Gkaragkounis et al., 2019; Sun et al., 2020), or noisy gradient estimates (Such et al., 2017). As a result, this can lead to uncoordinated suboptimal learning behavior because the actor cannot leverage the global information in the critic well. The issue is exacerbated more in the multi-agent settings due to the exponentially-sized joint action space (Yang et al., 2021) as well as the nature of the setting that requires *each* of the agent to learn a good policy for a successful joint policy. For example, in a basketball game, where there are two competing teams each consisting of five players. When one of the players passes the ball among them, it is important for all teammates to perform their duties well in their roles to win the game. As a result, if one of the agents in the team fails to learn a good policy, it can fail to cooperate with other agents for coordinated behaviors and lose the ball.

In this paper, we propose a surprisingly simple yet effective method for offline multi-agent continuous control, Offline MARL with Actor Rectification (OMAR), to better leverage the conservative value function via an effective combination of first-order policy gradient and zeroth-order optimization methods. Towards this goal, we add a regularizer to the actor loss, which encourages the actor to mimic actions from the zeroth-order optimizer that maximizes Q-values so that we can combine the best of both first-order policy gradient and zeroth-order optimization. The sampling mechanism is motivated by evolution strategies (Such et al., 2017; Conti et al., 2017; Mania et al., 2018), which recently emerged as another paradigm for solving sequential decision making tasks (Salimans et al., 2017). Specifically, the zeroth-order optimization part maintains an iteratively updated and refined Gaussian distribution to find better actions based on Q-values. Then, we rectify the policy towards this action to better leverage the conservative value function. We conduct extensive experiments in standard continuous control multi-agent particle environments and the complex multi-agent locomotion task to demonstrate its effectiveness. On all the benchmark tasks, OMAR outperforms the multi-agent version of offline RL algorithms including CQL (Kumar et al., 2020) and TD3+BC (Fujimoto & Gu, 2021), as well as a recent offline MARL algorithm MA-ICQ (Yang et al., 2021), and achieves the state-of-the-art performance.

The main contribution of this work can be summarized as follows. We propose the OMAR algorithm that effectively leverages both first-order and zero-order optimization for solving offline MARL tasks. In addition, we theoretically prove that OMAR leads to safe policy improvement. Finally, extensive experimental results demonstrate the effectiveness of OMAR, which significantly outperforms strong baseline methods and achieves state-of-the-art performance in datasets with different qualities in both decentralized and centralized learning paradigms.

## 2 BACKGROUND

We consider the framework of partially observable Markov games (POMG) (Littman, 1994; Hu et al., 1998), which extends Markov decision processes to the multi-agent setting. A POMG with $N$ agents is defined by a set of global states $\mathcal{S}$, a set of actions $\mathcal{A}_1, \ldots, \mathcal{A}_N$, and a set of observations $\mathcal{O}_1, \ldots, \mathcal{O}_N$ for each agent. At each timestep, each agent $i$ receive an observation $o_i$ and chooses an action based on its policy $\pi_i$. The environment transits to the next state according to the state transition function $\mathcal{P} : \mathcal{S} \times \mathcal{A}_1 \times \ldots \times \mathcal{A}_N \times \mathcal{S} \rightarrow [0, 1]$. Each agent receives a reward based on the reward function $r_i : \mathcal{S} \times \mathcal{A}_1 \ldots \times \mathcal{A}_N \rightarrow \mathbb{R}$ and a private observation $o_i : \mathcal{S} \rightarrow \mathcal{O}_i$. The initial state distribution is defined by $\rho : \mathcal{S} \rightarrow [0, 1]$. The goal is to find a set of optimal policies $\boldsymbol{\pi} = \{\pi_1, \ldots, \pi_N\}$, where each agent aims to maximize its own discounted return $\sum_{t=0}^{\infty} \gamma^t r_i^t$ with $\gamma$ denoting the discount factor. In the offline setting, agents learn from a fixed dataset $\mathcal{D}$ generated from the behavior policy $\boldsymbol{\pi}_\beta$ without interaction with the environments.

### 2.1 MULTI-AGENT ACTOR CRITIC

**Centralized critic.** Lowe et al. (2017) propose Multi-Agent Deep Deterministic Policy Gradients (MADDPG) under the centralized training with decentralized execution (CTDE) paradigm by extending the DDPG algorithm (Lillicrap et al., 2016) to the multi-agent setting. In CTDE, agents are trained in a centralized way where they can access to extra global information dur-

ing training while they need to learn decentralized policies in order to act based only on local observations during execution. In MADDPG, for an agent $i$, the centralized critic $Q_i$ is parameterized by $\theta_i$. It takes the global state action joint action as inputs, and aims to minimize the temporal difference error defined by $\mathcal{L}(\theta_i) = \mathbb{E}_{\mathcal{D}}\left[(Q_i(s, a_1, \ldots, a_n) - y_i)^2\right]$, where $y_i = r_i + \gamma \bar{Q}_i(s', a'_1, \cdots, a'_n)|_{a'_j = \bar{\pi}_j(o'_j)}$ and $\bar{Q}_i$ and $\bar{\pi}_i$ denote target networks. To reduce the overestimation problem in MADDPG, MATD3 (Ackermann et al., 2019) estimates the target value using double estimators based on TD3 (Fujimoto et al., 2018), where $y_i = r_i + \gamma \min_{k=1,2} \bar{Q}_i^k(s', a'_1, \cdots, a'_n)|_{a'_j = \bar{\pi}_j(o'_j)}$. Agents learn decentralized policies $\pi_i$ parameterized by $\phi_i$, which take only local observations as inputs, and are trained by multi-agent policy gradients according to $\nabla_{\phi_i} J(\pi_i) = \mathbb{E}_{\mathcal{D}}\left[\nabla_{\phi_i} \pi_i(a_i|o_i) \nabla_{a_i} Q_i(s, a_1, \ldots, a_n)|_{a_i = \pi_i(o_i)}\right]$, where $a_i$ is predicted from its policy while $a_{-i}$ are sampled from the replay buffer.

**Decentralized critic.** Although using centralized critics is widely-adopted in multi-agent actor-critic methods, it introduces scalability issues due to the exponentially sized joint action space w.r.t. the number of agents (Iqbal & Sha, 2019). On the other hand, independent learning approaches train decentralized critics that take only the local observation and action as inputs. It is shown in de Witt et al. (2020); Lyu et al. (2021) that decentralized value functions can result in more robust performance and be beneficial in practice compared with centralized critic approaches. de Witt et al. (2020) propose Independent Proximal Policy Optimization (IPPO) based on PPO (Schulman et al., 2017), and show that it can match or even outperform CTDE approaches in the challenging discrete control benchmark tasks (Samvelyan et al., 2019). We can also obtain the Independent TD3 (ITD3) algorithm based on decentralized critics, which is trained to minimize the temporal difference error defined by $\mathcal{L}(\theta_i) = \mathbb{E}_{\mathcal{D}}\left[(Q_i(o_i, a_i) - y_i)^2\right]$, where $y_i = r_i + \gamma \min_{k=1,2} \bar{Q}_i^k(o'_i, \bar{\pi}_i(o'_i))$.

## 2.2 Conservative Q-Learning

Conservative Q-Learning (CQL) (Kumar et al., 2020) adds a regularizer to the critic loss to address the extrapolation error and learns lower-bounded Q-values. It penalizes Q-values of state-action pairs sampled from a uniform distribution or a policy while encouraging Q-values for state-action pairs in the dataset to be large. Specifically, when built upon decentralized critic methods in MARL, the critic loss is defined as in Eq. (1), where $\alpha$ denotes the regularization coefficient and $\hat{\pi}_{\beta_i}$ is the empirical behavior policy of agent $i$.

$$\mathbb{E}_{\mathcal{D}_i}\left[(Q_i(o_i, a_i) - y_i)^2\right] + \alpha \mathbb{E}_{\mathcal{D}_i}\left[\log \sum_{a_i} \exp(Q_i(o_i, a_i)) - \mathbb{E}_{a_i \sim \hat{\pi}_{\beta_i}(a_i|o_i)}[Q_i(o_i, a_i)]\right] \quad (1)$$

## 3 Proposed Method

In this section, we first provide a motivating example where previous methods, such as CQL (Kumar et al., 2020) and TD3+BC (Fujimoto & Gu, 2021) can be inefficient in the face of the multi-agent setting. Then, we propose a method called Offline Multi-Agent Reinforcement Learning with Actor Rectification (OMAR), where we effectively combine first-order policy gradients and zeroth-order optimization methods for the actor to better optimize the conservative value function.

### 3.1 The Motivating Example

We design a Spread environment as shown in Figure 1(a) which involves $n$ agents and $n$ landmarks ($n \geq 1$) with 1-dimensional action space to demonstrate the problem and reveal interesting findings. For the multi-agent setting in the Spread task, $n$ agents need to learn how to cooperate to cover all landmarks and avoid colliding with each other or arriving at the same landmark by coordinating their actions. The experimental setup is the same as in Section 4.1.1.

Figure 1(b) demonstrates the performance of the multi-agent version of TD3+BC (Fujimoto & Gu, 2021), CQL (Kumar et al., 2020), and OMAR based on ITD3 in the medium-replay dataset from the two-agent Spread environment. As MA-TD3+BC is based on behavior regularization that compels the learned policy to stay close to the behavior policy, its performance largely depends on the quality of the dataset. Moreover, it can be detrimental to regularize policies to be close to the dataset in

multi-agent settings due to decentralized training and the resulting partial observations. MA-CQL outperforms MA-TD3+BC, which pushes down Q-values of state-action pairs that are sampled from a random or the current policy while pushing up Q-values for state-action pairs in the dataset.

Figure 1(c) demonstrates the performance improvement percentage of MA-CQL over the behavior policy with an increasing number of agents ranging from one to five. From Figure 1(c), we observe that its performance degrades dramatically as there are more agents.

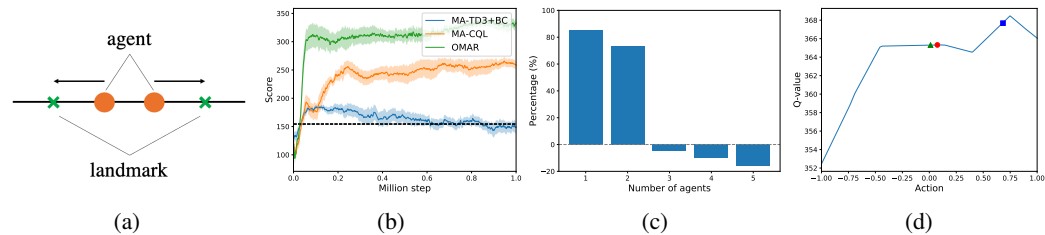

|      (a)      |      (b)      |      (c)      |      (d)      |

Figure 1: Analysis of MA-TD3+BC, MA-CQL, and OMAR in the medium-replay dataset from Spread. (a) Spread. (b) Performance. (c) Performance improvement percentage of MA-CQL over the behavior policy with varying number of agents. (d) Visualization of the Q-function landscape. The red circle represents the predicted action from the agent using MA-CQL. The green triangle and blue square represent the predicted action from the updated policy of MA-CQL and OMAR.

Towards mitigating the performance degradation, we identify a key issue in MA-CQL that solely regularizing the critic is insufficient for multiple agents to learn good policies for coordination. In Figure 1(d), we visualize the Q-function landscape of MA-CQL during training for an agent in a timestep, with the red circle corresponding to the predicted action from the actor. The green triangle represents the action predicted from the actor after the training step, where the policy gets stuck in a bad local optimum. The first-order policy gradient method is prone to local optima (Dauphin et al., 2014; Ahmed et al., 2019), where the agent can fail to globally leverage the conservative value function well and thus leading to suboptimal, uncoordinated learning behavior. Note that the problem is exacerbated more in the offline multi-agent setting due to the exponentially sized joint action space w.r.t. the number of agents (Yang et al., 2021). In addition, it usually requires *each* of the agent to learn a good policy for coordination to solve the task, and the suboptimal policy by any agent could result in uncoordinated global failure.

Tables 1 and 2 show the performance of MA-CQL by increasing the learning rate or the number of updates for the actor. The results illustrate that, to solve this challenging problem, we need a better solution than blindly tuning hyperparameters. In the next section, we introduce how we tackle this problem by combining zeroth-order optimization with current RL algorithms.

Table 1: Performance of MA-CQL with larger learning rate for the actor.

| Learning rate | $1e-2$ | $5e-2$ | $1e-1$ |
|---|---|---|---|
| Performance | 267.9 $\pm$19.0 | 202.0 $\pm$38.9 | 100.1 $\pm$36.4 |

Table 2: Performance of MA-CQL with larger number of updates for the actor.

| # Updates | 1 | 5 | 20 |
|---|---|---|---|
| Performance | 267.9 $\pm$19.0 | 278.6 $\pm$14.8 | 263.7 $\pm$23.1 |

## 3.2 Offline Multi-Agent Reinforcement Learning with Actor Rectification

Our key identification as above is that policy gradient improvements are prone to local optima given a bad value function landscape. It is important to note that this presents a particularly critical challenge in the multi-agent setting since it is sensitive to suboptimal actions. Zeroth-order optimization methods, *e.g.*, evolution strategies (Rubinstein & Kroese, 2013; Such et al., 2017; Conti et al., 2017; Salimans et al., 2017; Mania et al., 2018), offer an alternative for policy optimization and are also robust to local optima (Rubinstein & Kroese, 2013).

We propose Offline Multi-Agent Reinforcement Learning with Actor Rectification (OMAR) which incorporates sampled actions based on Q-values to rectify the actor so that it can escape from bad

local optima. For simplicity of presentation, we demonstrate our method based on the decentralized training paradigm introduced in Section 2.1. Note that it can also be applied to centralized critics, as shown in Section 4.1.4. Specifically, we add a regularizer to the policy objective:

$$\mathbb{E}_{\mathcal{D}_i}\left[(1-\tau)Q_i(o_i, \pi_i(o_i)) - \tau\left(\pi_i(o_i) - \hat{a}_i\right)^2\right] \tag{2}$$

where $\hat{a}_i$ is the action provided by the zeroth-order optimizer and $\tau \in [0,1]$ denotes the regularization coefficient. Note that TD3+BC (Fujimoto & Gu, 2021) uses the seen action in the dataset for $\hat{a}_i$. The distinction between *optimized* and *seen* actions enables OMAR to perform well even if the dataset quality is from mediocre to low.

We borrow intuition for sampling actions from recent evolution strategy (ES) algorithms, which show a welcoming avenue towards using zeroth-order method for policy optimization. For example, the cross-entropy method (CEM) (Rubinstein & Kroese, 2013), a popular ES algorithm, has shown great potential in RL (Lim et al., 2018), especially by sampling in the parameter space of the actor (Pourchot & Sigaud, 2019). However, CEM does not scale to tasks with high-dimensional space well (Nagabandi et al., 2020). We therefore propose to sample actions in a softer way motivated by Williams et al. (2015); Lowrey et al. (2018). Specifically, we sample actions according to an iteratively refined Gaussian distribution $\mathcal{N}(\mu_i, \sigma_i)$. At each iteration $j$, we draw $K$ candidate actions by $a_i^j \sim \mathcal{N}(\mu_i^j, \sigma_i^j)$ and evaluate their Q-values. The mean and standard deviation of the sampling distribution is updated and refined by Eq. (3), which produces a softer update and leverages more samples in the update (Nagabandi et al., 2020). The OMAR algorithm is shown in Algorithm 1.

$$\mu_i^{j+1} = \frac{\sum_{k=1}^K \exp(\beta Q_i^k)a_i^k}{\sum_{m=1}^K \exp(\beta Q_i^m)}, \quad \sigma_i^{j+1} = \sqrt{\sum_{k=1}^K \left(a_i^k - \mu_i^j\right)^2}. \tag{3}$$

Besides the algorithmic design, we also prove that OMAR gives a safe policy improvement guarantee. Let $J(\pi_i)$ denote the discounted return of a policy $\pi_i$ in the empirical MDP $\hat{M}_i$ which is induced by transitions in the dataset $\mathcal{D}_i$, i.e., $\hat{M}_i = \{(o_i, a_i, r_i, o_i') \in \mathcal{D}_i\}$. In Theorem 1, we give a lower bound on the difference between the policy performance of OMAR over the empirical behavior policy $\hat{\pi}_{\beta_i}$ in the empirical MDP $\hat{M}_i$. The proof can be found in Appendix A.

**Theorem 1.** *Let $\pi_i^*$ be the policy obtained by optimizing Eq. (2). Then, we have that* $J(\pi_i^*) - J(\hat{\pi}_{\beta_i}) \geq \frac{\alpha}{1-\gamma}\mathbb{E}_{o_i \sim d^{\pi_i^*}(o_i)}[D(\pi_i^*, \hat{\pi}_{\beta_i})(o_i)] + \frac{\tau}{1-\tau}\mathbb{E}_{o_i \sim d^{\pi_i^*}(o_i)}\left[(\pi_i^*(o_i) - \hat{a}_i)^2\right] - \frac{\tau}{1-\tau}\mathbb{E}_{o_i \sim d^{\hat{\pi}_{\beta_i}}(o_i), a_i \sim \hat{\pi}_{\beta_i}}\left[(a_i - \hat{a}_i)^2\right],$ *where* $D(\pi_i, \hat{\pi}_{\beta_i})(o_i) = \frac{1}{\hat{\pi}_{\beta_i}(\pi_i(o_i)|o_i)} - 1$, *and* $d^{\pi_i}(o_i)$ *is the marginal discounted distribution of observations of policy $\pi_i$.*

As shown in Theorem 1, the difference between the second and third terms on the right-hand side is the difference between two expected distances. The former corresponds to the gap between the optimal action and the action from our zeroth-order optimizer, while the latter corresponds to the gap between the action from the behavior policy and the optimized action. Since both terms can be bounded, we find that OMAR gives a safe policy improvement guarantee over $\hat{\pi}_{\beta_i}$.

**Discussion of the effect of OMAR in the Spread environment.** We now investigate whether OMAR can address the identified problem and analyze its effect in the Spread environment introduced in Section 3.1. In Figure 1(d), the blue square corresponds to the action from the updated actor using OMAR according to Eq. (2). In contrast to the policy update in MA-CQL, OMAR can better leverage the global information in the critic and help the actor to escape from the bad local optima. Figure 1(b) further validates that OMAR significantly improves MA-CQL in terms of both performance and efficiency. Figure 2 shows the performance improvement percentage of OMAR over MA-CQL with varying number of agents, where OMAR always outperforms MA-CQL. We also notice that the performance improvement of OMAR over MA-CQL is much more significant in the multi-agent setting in the Spread task than the

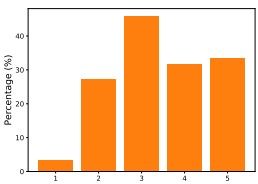

Figure 2: Performance improvement percentage of OMAR over MA-CQL with varying number of agents.

single-agent setting, which echoes with what is discussed above that the problem becomes more critical in scenarios with more agents that requires each of the agents to learn a good policy to cooperate for solving the task.

---

**Algorithm 1** Offline Multi-Agent Reinforcement Learning with Actor Rectification (OMAR).

---

1: Initialize $Q$-networks $Q_i^1$, $Q_i^2$ and policy networks $\pi_i$ with random parameters $\theta_1^i$, $\theta_2^i$, $\phi_i$, and
   target networks with $\bar{\theta}_i^1 \leftarrow \theta_i^1$, $\bar{\theta}_i^2 \leftarrow \theta_i^2$, and $\bar{\phi}_i \leftarrow \phi_i$ for each agent $i \in [1, N]$
2: **for** training step $t = 1$ to $T$ **do**
3:     **for** agent $i = 1$ to $N$ **do**
4:         Sample a random minibatch of $S$ samples $(o_i, a_i, r_i, o_i')$ from $\mathcal{B}$
5:         Set $y = r_i + \gamma \min \left( \bar{Q}_i^1(o_i', \pi_i(o_i' + \epsilon)), \bar{Q}_i^2(o_i', \pi_i(o_i' + \epsilon)) \right)$
6:         Update critics $\theta_i$ to minimize Eq. (1)
7:         Initialize $\mathcal{N}(\mu_i, \sigma_i)$
8:         **for** iteration $j = 1$ to $J$ **do**
9:             Draw a population with $K$ individuals $\hat{\mathcal{A}}_i = \{\hat{a}_i^k \sim \mathcal{N}(\mu_i, \sigma_i)\}_{k=1}^K$
10:           Estimate $Q$-values for $K$ individuals in the population $\{Q_i^1(o_i, \hat{a}_i^k)\}_{k=1}^K$
11:           Update $\mu_i$ and $\sigma_i$ of the distribution according to Eq. (3)
12:         Obtain the picked candidate action $\hat{a}_i = \arg\max_{\hat{a}_i \in \hat{\mathcal{A}}_i \cup \pi_i(o_i)} Q_i^1(o_i, \hat{a}_i)$
13:         Update the actor $\phi_i \leftarrow \max_{\phi_i} \frac{1}{S} \sum (1 - \tau) Q_i^1(o_i, \pi_i(o_i)) - \tau \left( \pi_i(o_i) - \hat{a}_i \right)^2$
14:         Update target networks: $\bar{\theta}_i^j \leftarrow \rho\theta_i^j + (1 - \rho)\bar{\theta}_i^j$ and $\bar{\phi}_i \leftarrow \rho\phi_i + (1 - \rho)\bar{\phi}_i$

---

## 4 EXPERIMENTS

In this section, we conduct a series of experiments to study the following key questions: i) How does OMAR compare against state-of-the-art offline RL and offline MARL methods? ii) What is the effect of critical hyperparameters and the sampling scheme? iii) Does the method help in both centralized training and decentralized training paradigms? iv) Can OMAR scale to the more complex continuous multi-agent locomotion tasks?

### 4.1 MULTI-AGENT PARTICLE ENVIRONMENTS

#### 4.1.1 EXPERIMENTAL SETUP

We first conduct a series of experiments in the widely-adopted multi-agent particle tasks (Lowe et al., 2017) as shown in Figure 5 in Appendix B.1. The cooperative navigation task includes 3 agents and 3 landmarks, where agents are rewarded based on the distance to the landmarks and penalized for colliding with each other. Thus, it is important for agents to cooperate to cover all landmarks without collision. In predator-prey, 3 predators aim to catch the prey. The predators need to cooperate to surround and catch the prey as the predators are slower than the prey. The world task involves 4 slower cooperating agents that aim to catch 2 faster adversaries, where adversaries desire to eat foods while avoiding being captured.

We construct a variety of datasets according to behavior policies with different qualities based on adding noises to the MATD3 algorithm to increase diversity following previous work (Fu et al., 2020). The random dataset is generated by rolling out a randomly initialized policy for 1 million (M) steps. We obtain the medium-replay dataset by recording all samples in the experience replay buffer during the training process until the policy reached the medium level of performance. The medium dataset consists of 1M samples by unrolling a partially-pretrained policy in the online setting whose performance reaches a medium level of the performance. The expert dataset is constructed by 1M expert demonstrations from an online fully-trained policy.

We compare OMAR against state-of-the-art offline RL algorithms including CQL (Kumar et al., 2020) and TD3+BC (Fujimoto & Gu, 2021). We also compare with a recent offline MARL algorithm MA-ICQ (Yang et al., 2021). We build all methods on independent TD3 based on decentralized critics following de Witt et al. (2020), while we also consider centralized critics based on MATD3 following Yu et al. (2021) in Section 4.1.4. All baselines are implemented based on the open-source code[1]. Each algorithm is run for five random seeds, and we report the mean performance with standard deviation. A detailed description of the construction of the datasets and hyperparameters can be found in Appendix B.1.

---

[1] `https://github.com/shariqiqbal2810/maddpg-pytorch`

### 4.1.2 PERFORMANCE COMPARISON

Table 3 summarizes the average normalized scores in different datasets in multi-agent particle environments, where the learning curves are shown in Appendix B.2. The normalized score is computed as $100 \times (S - S_{\text{random}})/(S_{\text{expert}} - S_{\text{random}})$ following Fu et al. (2020) As shown, the performance of MA-TD3+BC highly depends on the quality of the dataset. As the MA-ICQ method is based on only trusting seen state-action pairs in the dataset, it does not perform well in datasets with more diverse data distribution including random and medium-replay datasets, while generally matches the performance of MA-TD3+BC in datasets with more narrow distribution including medium and expert. MA-CQL matches or outperforms MA-TD3+BC in datasets with lower quality except for the expert dataset, as it does not rely on constraining the learning policy to stay close to the behavior policy. Our OMAR method significantly outperforms all baseline methods and achieves state-of-the-art performance. We attribute the performance gain to the actor rectification scheme that is independent of data quality and improves global optimization. In addition, OMAR does not incur much computation cost and only takes $4.7\%$ more runtime on average compared with that of MA-CQL.

Table 3: Averaged normalized score of OMAR and baselines in multi-agent particle environments.

| | | MA-ICQ | MA-TD3+BC | MA-CQL | OMAR |
|---|---|---|---|---|---|
| Random | Cooperative navigation | $6.3 \pm 3.5$ | $9.8 \pm 4.9$ | $24.0 \pm 9.8$ | $\mathbf{34.4} \pm 5.3$ |
| | Predator-prey | $2.2 \pm 2.6$ | $5.7 \pm 3.5$ | $5.0 \pm 8.2$ | $\mathbf{11.1} \pm 2.8$ |
| | World | $1.0 \pm 3.2$ | $2.8 \pm 5.5$ | $0.6 \pm 2.0$ | $\mathbf{5.9} \pm 5.2$ |
| Medium-replay | Cooperative navigation | $13.6 \pm 5.7$ | $15.4 \pm 5.6$ | $20.0 \pm 8.4$ | $\mathbf{37.9} \pm 12.3$ |
| | Predator-prey | $34.5 \pm 27.8$ | $28.7 \pm 20.9$ | $24.8 \pm 17.3$ | $\mathbf{47.1} \pm 15.3$ |
| | World | $12.0 \pm 9.1$ | $17.4 \pm 8.1$ | $29.6 \pm 13.8$ | $\mathbf{42.9} \pm 19.5$ |
| Medium | Cooperative navigation | $29.3 \pm 5.5$ | $29.3 \pm 4.8$ | $34.1 \pm 7.2$ | $\mathbf{47.9} \pm 18.9$ |
| | Predator-prey | $63.3 \pm 20.0$ | $65.1 \pm 29.5$ | $61.7 \pm 23.1$ | $\mathbf{66.7} \pm 23.2$ |
| | World | $71.9 \pm 20.0$ | $73.4 \pm 9.3$ | $58.6 \pm 11.2$ | $\mathbf{74.6} \pm 11.5$ |
| Expert | Cooperative navigation | $104.0 \pm 3.4$ | $108.3 \pm 3.3$ | $98.2 \pm 5.2$ | $\mathbf{114.9} \pm 2.6$ |
| | Predator-prey | $113.0 \pm 14.4$ | $115.2 \pm 12.5$ | $93.9 \pm 14.0$ | $\mathbf{116.2} \pm 19.8$ |
| | World | $109.5 \pm 22.8$ | $110.3 \pm 21.3$ | $71.9 \pm 28.1$ | $\mathbf{110.4} \pm 25.7$ |

### 4.1.3 ABLATION STUDY

**The effect of the regularization coefficient.** We first investigate the effect of the regularization coefficient $\tau$ in the actor loss in Eq. (2). Figure 3 shows the averaged normalized score of OMAR over different tasks with different values of $\tau$ in each kind of dataset. As shown, the performance of OMAR is sensitive to this hyperparameter, which controls the exploitation level of the critic. We find the best value of $\tau$ is neither close to 1 nor 0, showing that it is the combination of both policy gradients and the actor rectification that performs well. We also notice that the optimal value of $\tau$ is smaller for datasets with lower quality and more diverse data distribution including random and medium-replay, but larger for medium and expert datasets. In addition, the performance of OMAR with all values of $\tau$ matches or outperforms that of MA-CQL. This is the only hyperparameter that needs to be tuned in OMAR beyond MA-CQL.

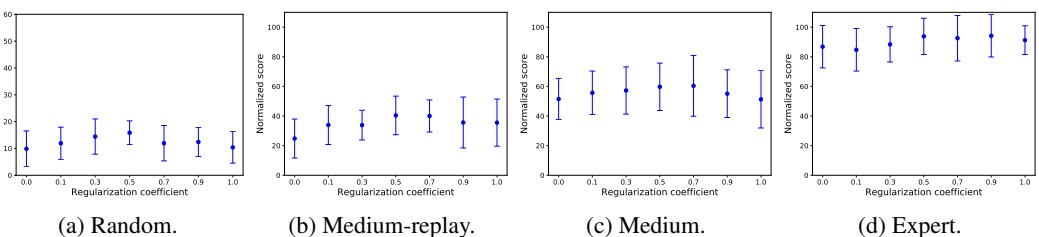

| (a) Random. | (b) Medium-replay. | (c) Medium. | (d) Expert. |

Figure 3: Ablation study on the effect of the regularization coefficient in different types of datasets.

**The effect of key hyperparameters in the sampling scheme.** Core hyperparameters for our sampling mechanism involves the number of iterations, the number of sampled actions, and the initial mean and standard deviation of the Gaussian distribution. Figures 4(a)-(d) show the performance comparison of OMAR with different values of these hyperparameters in the cooperative navigation task, where the grey dotted line corresponds to the normalized score of MA-CQL. As shown, our sampling mechanism is not sensitive to these hyperparameters, and we fix them to be the set with the best performance.

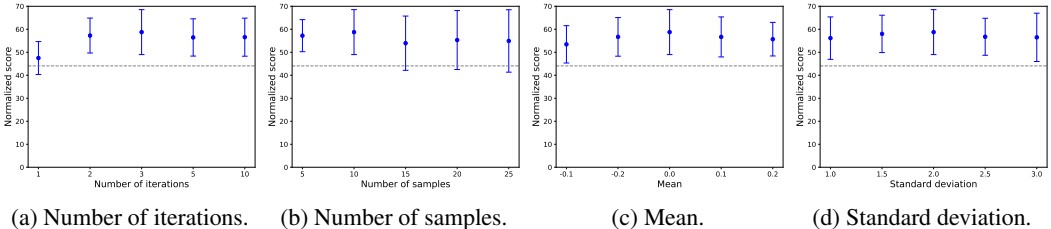

(a) Number of iterations.  (b) Number of samples.  (c) Mean.  (d) Standard deviation.

Figure 4: Ablation study on the effect of key hyperparameters in the sampling mechanism averaged over different types of datasets.

**The effect of the sampling mechanism.** We now analyze the effect of the zeroth-order optimization methods in OMAR, and compare it against random shooting and the cross-entropy method (CEM) (De Boer et al., 2005) in the cooperative navigation task. As shown in Table 4, our sampling mechanism significantly outperforms the random sampling scheme and CEM, with a larger margin in datasets with lower quality including random and medium-replay. The proposed sampling technique incorporates more samples into the distribution updates more effectively.

Table 4: Ablation study of OMAR with different sampling mechanisms in different types of datasets.

|  | Random | Medium-replay | Medium | Expert |
|---|---|---|---|---|
| OMAR (random) | $24.3 \pm 7.0$ | $23.5 \pm 5.3$ | $41.2 \pm 11.1$ | $101.0 \pm 5.2$ |
| OMAR (CEM) | $25.8 \pm 7.3$ | $32.6 \pm 5.1$ | $45.0 \pm 13.3$ | $106.4 \pm 13.8$ |
| OMAR | $\mathbf{34.4} \pm 5.3$ | $\mathbf{37.9} \pm 12.3$ | $\mathbf{47.9} \pm 18.9$ | $\mathbf{114.9} \pm 2.6$ |

### 4.1.4 APPLICABILITY ON CENTRALIZED TRAINING WITH DECENTRALIZED EXECUTION

In this section, we demonstrate the versatility of the method and show that it can also be applied and beneficial to methods based on centralized critics under the CTDE paradigm. Specifically, all baseline methods are built upon the MATD3 algorithm (Ackermann et al., 2019) using centralized critics as detailed in Section 2.1. Table 5 summarizes the averaged normalized score of different algorithms in each kind of dataset. As shown, OMAR (centralized) also significantly outperforms MA-ICQ (centralized) and MA-CQL (centralized), and matches the performance of MA-TD3+BC (centralized) in the expert dataset while outperforming it in other datasets.

Table 5: The average normalized score of different methods based on MATD3 with centralized critics under the CTDE paradigm.

|  | Random | Medium-reply | Medium | Expert |
|---|---|---|---|---|
| MA-ICQ | $5.2 \pm 5.5$ | $10.1 \pm 4.6$ | $27.4 \pm 5.3$ | $96.7 \pm 4.1$ |
| MA-TD3+BC | $7.9 \pm 2.2$ | $9.3 \pm 9.1$ | $29.4 \pm 3.7$ | $\mathbf{108.1} \pm 3.3$ |
| MA-CQL | $12.8 \pm 4.9$ | $11.2 \pm 6.6$ | $26.3 \pm 13.3$ | $69.5 \pm 15.7$ |
| OMAR | $\mathbf{21.6} \pm 4.6$ | $\mathbf{19.1} \pm 9.2$ | $\mathbf{33.7} \pm 14.5$ | $105.9 \pm 3.6$ |

## 4.2 MULTI-AGENT MUJOCO

In this section, we investigate whether OMAR can scale to more complex continuous control multi-agent tasks. Peng et al. (2020) introduce multi-agent locomotion tasks which extends the high-

dimensional MuJoCo locomotion tasks in the single-agent setting to the multi-agent case. We consider the two-agent HalfCheetah task (Kim et al., 2021) as shown in Appendix B.1, where the first and second agents control different parts of joints of the robot. Agents need to cooperate to make the robot run forward by coordinating the actions. We also construct different types of datasets following Fu et al. (2020) the same as in Section 4.1.1. Table 6 summarizes the average normalized scores in each kind of dataset in multi-agent HalfCheetah. As shown, OMAR significantly outperforms baseline methods in random, medium-replay, and medium datasets, and matches the performance of MA-TD3+BC in expert, demonstrating its effectiveness to scale to more complex control tasks.

Table 6: Average normalized score of different methods in multi-agent HalfCheetah.

|  | Random | Medium-reply | Medium | Expert |
|---|---|---|---|---|
| MA-ICQ | $7.4 \pm 0.0$ | $35.6 \pm 2.7$ | $73.6 \pm 5.0$ | $110.6 \pm 3.3$ |
| MA-TD3+BC | $7.4 \pm 0.0$ | $27.1 \pm 5.5$ | $75.5 \pm 3.7$ | $\mathbf{114.4} \pm 3.8$ |
| MA-CQL | $7.4 \pm 0.0$ | $41.2 \pm 10.1$ | $50.4 \pm 10.8$ | $64.2 \pm 24.9$ |
| OMAR | $\mathbf{15.4} \pm 12.3$ | $\mathbf{57.7} \pm 5.1$ | $\mathbf{80.7} \pm 10.2$ | $\mathbf{113.5} \pm 4.3$ |

## 5 RELATED WORK

**Offline reinforcement learning.** Many recent papers achieve improvements in offline RL (Wu et al., 2019; Kumar et al., 2019; Yu et al., 2020; Kidambi et al., 2020) that address the extrapolation error. Behavior regularization typically compels the learning policy to stay close to the behavior policy. Yet, its performance relies heavily on the quality of the dataset. Critic regularization approaches typically add a regularizer to the critic loss that pushes down Q-values for actions sampled from a given policy (Kumar et al., 2020). As discussed above, it can be difficult for the actor to best leverage the global information in the critic as policy gradient methods are prone to local optima, which is particularly important in the offline multi-agent setting.

**Multi-agent reinforcement learning.** A number of multi-agent policy gradient algorithms train agents based on centralized value functions (Lowe et al., 2017; Foerster et al., 2018; Yu et al., 2021) while another line of research focuses on decentralized training (de Witt et al., 2020). Yang et al. (2021) show that the extrapolation error in offline RL can be more severe in the multi-agent setting than the single-agent case due to the exponentially sized joint action space w.r.t. the number of agents. In addition, it presents a critical challenge in the decentralized setting when the datasets for each agent only consist of its own action instead of the joint action (Jiang & Lu, 2021). Jiang & Lu (2021) address the challenges based on the behavior regularization BCQ (Fujimoto et al., 2019) algorithm while Yang et al. (2021) propose to estimate the target value based on the next action from the dataset. As a result, both methods largely depend on the quality of the dataset.

**Zeroth-order optimization method.** It has been recently shown in (Such et al., 2017; Conti et al., 2017; Mania et al., 2018) that evolutionary strategies (ES) emerge as another paradigm for continuous control. Recent research shows that it is potential to combine RL with ES to reap the best of both worlds (Khadka & Tumer, 2018; Pourchot & Sigaud, 2019) in the high-dimensional parameter space for the actor. Sun et al. (2020) replace the policy gradient update via supervised learning based on sampled noises from random shooting. Kalashnikov et al. (2018); Lim et al. (2018); Simmons-Edler et al. (2019); Peng et al. (2020) extend Q-learning based approaches to handle continuous action space based on the popular cross-entropy method (CEM) in ES.

## 6 CONCLUSION

In this paper, we identify the problem that when extending conservatism-based RL algorithms to offline multi-agent scenarios, the performance degrades along increasing number of agents. To tackle this problem, propose Offline Multi-Agent RL with Actor Rectification (OMAR) that combines first-order policy gradient with zeroth-order optimization. We find that OMAR can successfully help the actor escape from bad local optima and consequently find better actions. OMAR achieves state-of-the-art performance on multi-agent continuous control tasks empirically.

## REPRODUCIBILITY STATEMENT

We include all details for our experiments in Appendix B.1 including open-source implementations of environments and algorithms, with a detailed description for the hyperparameters and network structures. For Theorem 1, its proof can be found in Appendix A. The code will be open-sourced upon publication of the work at `https://sites.google.com/view/iclr2022omar`.

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

## A    PROOF OF THEOREM 1

**Theorem 1.**    *Let $\pi_i^*$ be the policy obtained by optimizing Eq.   (2).    Then, we have that $J(\pi_i^*) - J(\hat{\pi}_{\beta_i}) \geq \frac{\alpha}{1-\gamma}\mathbb{E}_{o_i \sim d^{\pi_i^*}(o_i)}[D(\pi_i^*, \hat{\pi}_{\beta_i})(o_i)] + \frac{\tau}{1-\tau}\mathbb{E}_{o_i \sim d^{\pi_i^*}(o_i)}\left[(\pi_i^*(o_i) - \hat{a}_i)^2\right] - \frac{\tau}{1-\tau}\mathbb{E}_{o_i \sim d^{\hat{\pi}_{\beta_i}}(o_i), a_i \sim \hat{\pi}_{\beta_i}}\left[(a_i - \hat{a}_i)^2\right]$, where $D(\pi_i, \hat{\pi}_{\beta_i})(o_i) = \frac{1}{\hat{\pi}_{\beta_i}(\pi_i(o_i)|o_i)} - 1$, and $d^{\pi_i}(o_i)$ is the marginal discounted distribution of observations of policy $\pi_i$.*

*Proof.*  For OMAR, we have the following iterative update for agent $i$:

$$\hat{Q}_i^{k+1} \leftarrow \arg\min_{Q_i} \alpha\mathbb{E}_{o_i \sim \mathcal{D}_i}\left[\mathbb{E}_{a_i \sim \tilde{\pi}_i(a_i|o_i)}[Q_i(o_i, a_i)] - \mathbb{E}_{a_i \sim \hat{\pi}_{\beta_i}(a_i|o_i)}[Q_i(o_i, a_i)]\right]$$
$$+ \frac{1}{2}\mathbb{E}_{o_i, a_i, o_i' \sim \mathcal{D}}\left[\left(Q_i(o_i, a_i) - \hat{\mathcal{B}}^{\pi_i}\hat{Q}_i^k(o_i, a_i)\right)^2\right], \quad (4)$$

where $\tilde{\pi}_i(a_i|o_i) = 1$ if and only if $a_i = \pi_i(o_i)$.

Let $\hat{Q}_i^{k+1}$ be the fixed point of solving Equation (4) by setting the derivative of Eq. (4) with respect to $Q_i$ to be 0, then we have that

$$\hat{Q}_i^{k+1}(o_i, a_i) = \hat{\mathcal{B}}^{\pi_i}\hat{Q}_i^k(o_i, a_i) - \alpha\left(\frac{I_{a_i = \pi_i(o_i)}}{\hat{\pi}_{\beta_i}(a_i|o_i)} - 1\right), \quad (5)$$

where $I$ is the indicator function.

Denote $D(\pi_i, \hat{\pi}_{\beta_i})(o_i) = \frac{1}{\hat{\pi}_{\beta_i}(\pi_i(o_i)|o_i)} - 1$, and we obtain the difference between the value function $\hat{V}_i(o_i)$ and the original value function as:

$$\hat{V}_i(o_i) = V_i(o_i) - \alpha D(\pi_i, \hat{\pi}_{\beta_i})(o_i), \quad (6)$$

Then, the policy that minimizes the loss function defined in Eq. (2) is equivalently obtained by maximizing

$$(1 - \tau)\left(J(\pi_i) - \alpha\frac{1}{1-\gamma}\mathbb{E}_{o_i \sim d^{\pi_i}_{\hat{M}_i}(o_i)}[D(\pi_i, \hat{\pi}_{\beta_i})(o_i)]\right) - \tau\mathbb{E}_{o_i \sim d^{\pi_i}_{\hat{M}_i}(o_i)}\left[(\pi_i(o_i) - \hat{a}_i)^2\right]. \quad (7)$$

Therefore, we obtain that

$$(1 - \tau)\left(J(\pi_i^*) - \alpha\frac{1}{1-\gamma}\mathbb{E}_{o_i \sim d^{\pi_i^*}_{\hat{M}_i}(o_i)}[D(\pi_i^*, \hat{\pi}_{\beta_i})(o_i)]\right) - \tau\mathbb{E}_{o_i \sim d^{\pi_i^*}_{\hat{M}_i}(o_i)}\left[(\pi_i^*(o_i) - \hat{a}_i)^2\right]$$
$$\geq (1 - \tau)J(\hat{\pi}_{\beta_i}) - \tau\mathbb{E}_{o_i \sim d^{\hat{\pi}_{\beta_i}}_{\hat{M}_i}(o_i), a_i \sim \hat{\pi}_{\beta_i}(a_i|o_i)}\left[(a_i - \hat{a}_i)^2\right]. \quad (8)$$

Then, from Eq. (8) we obtain the result.    $\square$

## B    MORE DETAILS OF THE EXPERIMENTS

### B.1    EXPERIMENTAL SETUP

**Tasks.**  We adopt the open-source implementations for multi-agent particle environments[2] from (Lowe et al., 2017) and Multi-Agent MuJoCo[3] from (Peng et al., 2020). Figure 5 illustrates the tasks. The expert and random scores for cooperative navigation, predator-prey, world, and two-agent HalfCheetah are $\{516.8, 159.8\}$, $\{185.6, -4.1\}$, $\{79.5, -6.8\}$, and $\{3568.8, -284.0\}$, respectively.

---

[2]https://github.com/openai/multiagent-particle-envs
[3]https://github.com/schroederdewitt/multiagent_mujoco

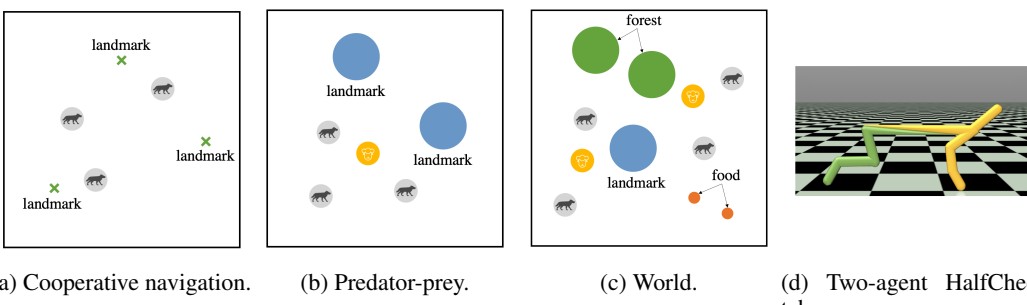

(a) Cooperative navigation.    (b) Predator-prey.    (c) World.    (d) Two-agent HalfCheetah.

Figure 5: Multi-agent particle environments and Multi-Agent HalfCheetah.

**Baselines.** All baseline methods are implemented based on an open-source implementation[4] from (Iqbal & Sha, 2019), where we implement MA-TD3+BC[5], MA-CQL[6], and MA-ICQ[7] based on authors' open-source implementations with fine-tuned hyperparameters. For MA-CQL, we tune a best critic regularization coefficient from $\{0.1, 0.5, 1.0, 5.0\}$ following (Kumar et al., 2020) for each task. Specifically, we use the discount factor $\gamma$ of $0.99$. We sample a minibatch of $1024$ samples from the dataset for updating each agent's actor and critic using the Adam (Kingma & Ba, 2014) optimizer with the learning rate to be $0.01$. The target networks for the actor and critic are soft updated with the update rate to be $0.01$. Both the actor and critic networks are feedforward networks consisting of two hidden layers with $64$ neurons per layer using ReLU activation. For OMAR, the only hyperparameter that requires tuning is the regularization coefficient $\lambda$, where we use a smaller value for datasets with more diverse data distribution in random and medium-replay with a value of $0.5$, while we use a larger value for datasets with more narrow data distribution in medium and expert with values of $0.7$ and $0.9$ respectively. As OMAR is insensitive to the hyperparameters of the sampling mechanism, we set them to a fixed set of values for all types of datasets in all tasks, where the number of iteration is $3$, the number of samples is $10$, the mean is $0.0$, and the standard deviation is $2.0$. The code will be released upon publication of the paper.

---

[4]https://github.com/shariqiqbal2810/maddpg-pytorch
[5]https://github.com/sfujim/TD3_BC
[6]https://github.com/aviralkumar2907/CQL
[7]https://github.com/YiqinYang/ICQ

## B.2 LEARNING CURVES

Figure 6 demonstrates the learning curves of MA-ICQ, MA-TD3+BC, MA-CQL and OMAR in different types of datasets in multi-agent particle environments, where the solid line and shaded region represent mean and standard deviation, respectively.

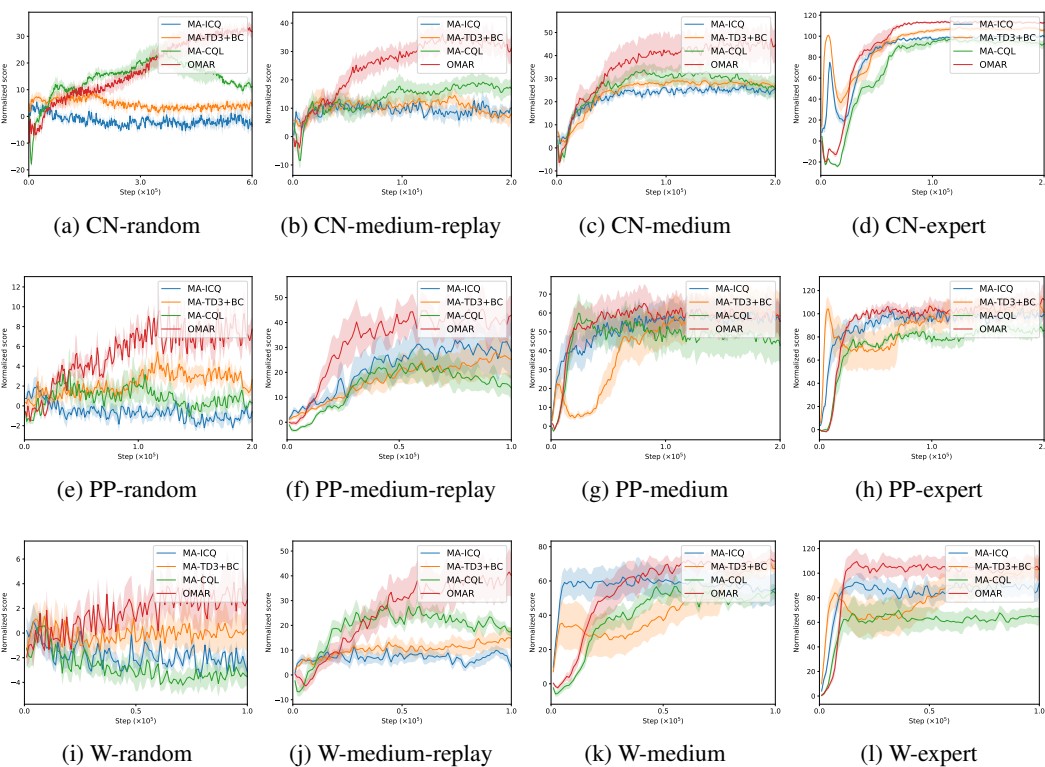

Figure 6: Learning curves of MA-ICQ, MA-TD3+BC, MA-CQL, and OMAR in multi-agent particle environments (CN, PP, and W is abbreviated for cooperative navigation, predator-prey, and world respectively).

## B.3 STARCRAFT II MICROMANAGEMENT BENCHMARK

**Setup.** We investigate the effectiveness of OMAR in larger-scale tasks based on the challenging StarCraft II micromanagement benchmark (Samvelyan et al., 2019) on maps with an increasing number of agents and difficulties including 2s3z, 3s5z, 1c3s5z, and 2c_vs_64zg. Details for the tasks are shown in Table 7. We compare OMAR and MA-CQL based on the evaluation protocol in Kumar et al. (2020); Agarwal et al. (2020); Gulcehre et al. (2020), where datasets are constructed following Agarwal et al. (2020); Gulcehre et al. (2020) by recording samples observed during training. Each dataset consists of 1 million samples. We use the Gumbel-Softmax reparameterization trick (Jang et al., 2016) to generate discrete actions for MATD3 since it requires differentiable policies (Lowe et al., 2017; Iqbal & Sha, 2019; Peng et al., 2020). All implementations are based on open-sourced code [8] and the same experimental setup as in Appendix B.1.

---

[8] https://github.com/oxwhirl/comix

Table 7: Specs of tested maps in the StarCraft II micromanagement benchmark.

| Name | Agents | Enemies |
|---|---|---|
| 2s3z | 2 Stalkers and 3 Zealots | 2 Stalkers and 3 Zealots |
| 3s5z | 3 Stalkers and 5 Zealots | 3 Stalkers and 5 Zealots |
| 1c3s5z | 1 Colossi, 3 Stalkers and 5 Zealots | 1 Colossi, 3 Stalkers and 5 Zealots |
| 2c_vs_64zg | 2 Colossi | 64 Zerglings |

**Results.** Figure 7 demonstrates the comparison result in test win rates. As shown, OMAR significantly outperforms MA-CQL in performance and learning efficiency, and the average performance gain of OMAR compared to MA-CQL is 76.7% in all tested maps.

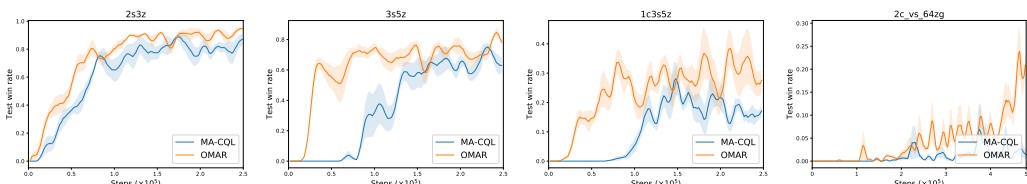

Figure 7: Comparison of OMAR and MA-CQL in StarCraft II micromanagement tasks.

## B.4 DISCUSSION ON OMAR IN ONLINE/OFFLINE, MULTI-AGENT/SINGLE-AGENT SETTINGS

We now investigate the effectiveness of OMAR in the four following settings: (1) online multi-agent setting, (2) online single-agent setting, (3) offline single-agent setting, and (4) offline multi-agent setting in the Spread environment shown in Figure 1(a).

**Setup.** For the online setting, we build our method upon the MATD3 algorithm with our proposed policy objective in Eq. (2), and evaluate the performance improvement percentage of our method over MATD3. The results for the online setting are shown in the right part in Figure 8, where the x-axis corresponds to the performance improvement percentage and the y-axis corresponds to the number of agents indicating whether its single-agent or multi-agent setting. We combine the results in Figure 2 for the offline setting which shows the performance improvement percentage of OMAR over MA-CQL in the left part in Figure 8 for a better understanding of the effectiveness of our method in different settings.

**Results.** As shown in Figure 8, our method is generally applicable in all the settings. However, the performance improvement is much more significant in the offline setting (left part) than the online case (right part), because the agents cannot explore and interact with the environment. Intuitively, in the online setting, if the actor has not well exploited the global information in the value function, it can still interact with the environment to collect better experiences for improving the estimation of the value function and provides a better guidance for the policy. However, no exploration and interaction with the environment for new data collection are allowed in the offline setting. Thus, it is much harder for an agent to escape from a bad local optimum and difficult for the actor to best leverage the global information in the critic. This presents an even more challenging problem in multi-agent RL because multiple agents result in an exponentially-sized joint action space as well as the nature of the setting that requires a coordinated joint policy. As expected, we also find that the performance gain is more significant in the offline multi-agent domain, which requires each of the agents to learn a good policy for a successful joint policy for coordination. Otherwise, it can lead to an uncoordinated global failure.

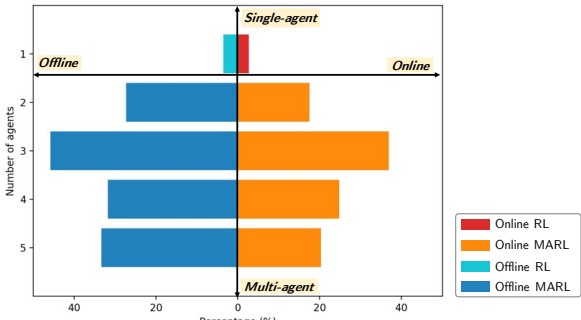

Figure 8: Performance improvement percentage of our method over MATD3 in the online (right part) setting and MA-CQL in the offline setting (left part) with a varying number of agents in the Spreak task. The first, second, thrid, and fourth quadrants correspond to the online RL, offline RL, offline multi-agent RL, and online multi-agent RL settings.

### B.5 THE EFFECT OF THE SIZE OF THE DATASET

In this section, we conduct an ablation study to investigate the effect of the size of the dataset following the experimental protocol in Agarwal et al. (2020). We first generate a full replay dataset by recording all samples in the replay buffer encountered during the training course for 1 million steps in the cooperative navigation task. Then, we randomly sample $N\%$ experiences from the full replay dataset and obtain several smaller datasets with the same data distribution, where $N \in \{0.1, 1, 10, 20, 50, 100\}$.

Figure 9 shows that the performance of MA-CQL increases given more data points for $N \in \{1, 10, 20\}$. However, it does not further increase given an even larger amount of data, which performs much worse than the fully-trained online agents and fails to recover their performance. On the contrary, OMAR always outperforms MA-CQL by a large margin when $N > 1\%$, whose performance is much closer to the fully-trained online agents given more data points. Therefore, the optimality issue still persists when dataset size becomes larger (e.g., it can take a very long time to escape from them if the objective contains very flat regions (Ahmed et al., 2019)). In addition, the zeroth-order optimizer part in OMAR can better guide the actor given a larger amount of data points with a more accurate value function.

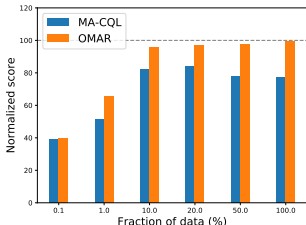

Figure 9: Normalized score of OMAR and MA-CQL trained using a fraction of the entire replay dataset in cooperative navigation.

### B.6 ADDITIONAL RESULTS OF OMAR IN SINGLE-AGENT ENVIRONMENTS

Besides the single-agent setting of the Spread task we have shown in Figure 2, we also evaluate the effectiveness of our method in single-agent tasks by comparing it with CQL in the Maze2D domain from the D4RL benchmark (Fu et al., 2020). Table 8 shows the results in an increasing order of complexity of the maze (maze2d-umaze, maze2d-medium, maze2d-large). Based on the results in Table 8 and Figure 2, we observe that OMAR performs much better than CQL, which indicates that OMAR is effective in the offline single-agent tasks.

Table 8: Averaged normalized score of OMAR and CQL in the single-agent Maze2D domain from D4RL.

|  | maze2d-umaze | maze2d-medium | maze2d-large |
|---|---|---|---|
| CQL | $109.8 \pm 23.9$ | $106.4 \pm 11.0$ | $94.6 \pm 44.6$ |
| OMAR | $\mathbf{124.7} \pm 7.6$ | $\mathbf{125.7} \pm 12.3$ | $\mathbf{157.7} \pm 12.3$ |

### B.7 ANALYSIS OF HOW COOPERATION AFFECT THE PERFORMANCE OF CQL IN MULTI-AGENT TASKS

We consider a non-cooperative version of the Spread task in Figure 1(a) which involves $n$ agents and $n$ landmarks, where each of the agents aims to navigate to its own unique target landmark. In contrast to the Spread task that requires cooperation, the reward function for each agent only depends on its distance to its target landmark. This is a variant of Spread that consists of multiple independent learning agents, and the performance is measured by the average return over all agents.

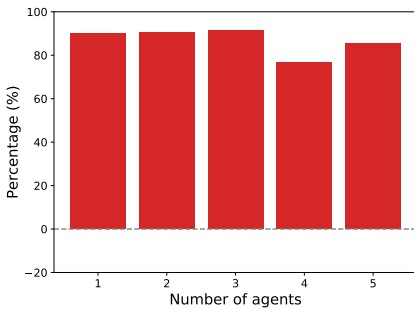

Figure 10: Performance improvement percentage of MA-CQL over the behavior policy with a varying number of agents in a non-cooperative version of the Spread task.

Figure 10 shows the result of the performance improvement percentage of MA-CQL over the behavior policy in the independent Spread task. As shown, the performance of CQL does not degrade with an increasing number of agents in this setting that does not require cooperation, unlike a dramatic performance decrease in the cooperative Spread task in Figure 1(c). The result further confirms that the issue we discovered is due to the failure of coordination.

### B.8 DISCUSSION ABOUT THE CENTRALIZED AND DECENTRALIZED CRITICS IN OFFLINE MULTI-AGENT RL

We attribute the lower performance in Table 5 (based on a centralized value function) compared to Table 3 (based on a decentralized value function) due to the base algorithm, where Table 9 shows the performance comparison of offline independent TD3 and offline multi-agent TD3 in different types of dataset in cooperative navigation. As shown, utilizing centralized critics underperforms decentralized critics in the offline setting. There has also been recent research (de Witt et al., 2020; Lyu et al., 2021) showing the benefits of decentralized value functions compared to a centralized one, which leads to a more robust performance. We attribute the performance loss of CTDE in the offline setting due to a more complex and higher-dimensional value function conditioning on all agent's actions and the global state that is harder to learn well without exploration.

Table 9: Averaged normalized score of ITD3 and MATD3 in cooperative navigation.

|  | Random | Medium-replay | Medium | Expert |
|---|---|---|---|---|
| ITD3 | $\mathbf{18.7} \pm 8.0$ | $\mathbf{19.9} \pm 4.7$ | $\mathbf{18.6} \pm 4.4$ | $\mathbf{75.5} \pm 7.9$ |
| MATD3 | $16.1 \pm 5.6$ | $12.7 \pm 6.1$ | $12.1 \pm 14.2$ | $1.6 \pm 2.7$ |

