# OpenReview forum: "Plan Better Amid Conservatism: Offline Multi-Agent Reinforcement Learning with Actor Rectification"
_ICLR.cc/2022/Conference — ICLR 2022 Submitted_

### Official Review · Reviewer_pt8q · 2021-11-02

**Correctness:** 3
**Technical Novelty And Significance:** 2
**Empirical Novelty And Significance:** 2
**Recommendation:** 5
**Confidence:** 3

**Main Review:**

The paper is overall clearly written. The intuition is discussed and the main idea is easy to follow with enough explanations. It is a bit surprising to see that such a simple combination of first-order and zeroth-order methods is able to deliver a good performance. The algorithm can be appealing given it seems quite easy to implement.

I do however have some comments:
(1) Theoretical: the authors state a theorem regarding the performance bound and argue that "OMAR gives a safe policy improvement guarantee". However, I do not see how "safe" the policy is. Can the authors provide more explanations about the bound? How large the lower bound could be? Is there easy example that one can get some concrete results? It's fine that the theorem is not strong since I consider this as primarily a practical paper, but some explanation is needed to avoid over-claiming the results..
(2) Practical: I think the empirical evaluation is not adequate. In particular, in the beginning, it is motivated that standard methods fail when there are more and more agents. The experiments, while confirming the efficacy of the proposed method, largely focus on cases where there are few agents.I appreciate the detailed ablation study the authors provide for different parameters, but I think a curve showing how the method compares as the number of agents increase to a large number is useful. The proposed method also doesn't seem to have scalability issue to run such an experiment.

**Summary Of The Paper:**

This work considers extending conservatism-based algorithms to offline RL with multi agents. The performance of standard algorithms often degrades significantly in this setting, especially when the number of agents increase. To resolve the issue, the authors propose a simple scheme which essentially combines first-order and zeroth-order policy optimization methods, with the goal to possibly extract the advantages of each method. Some empirical results demonstrate that the proposed algorithm can achieve better performance than standard baselines.


**Summary Of The Review:**

Overall, I think this paper contains some interesting and useful ideas. Some parts can be further improved as mentioned before and I hope that the author response will address them.

---

> ### Author Response · Authors · 2021-11-21
> **Author Response**
>
> We thank the reviewer for the thoughtful review and careful reading of our paper, and for noting that we "proposed interesting and useful ideas" and an "appealing algorithm with good performance", and the paper is "clearly written with enough explanations".
>
> *Q1:  Can authors provide more explanations about the bound in Theorem 1 regarding how safe the policy is? How large could the lower bound be? Is there an easy example where one can get some concrete results?*
>
> The first non-negative term on the RHS captures the policy performance increase due to the conservative Q-function. The second and third terms are two expected distances due to our introduced actor regularizer. Specifically, the former corresponds to the gap between the action from the zeroth-order optimizer $\hat{a}_i$ and $\pi_i^*(o_i)$ which is obtained by optimizing Eq. (2).
> The latter corresponds to the gap between $\hat{a}_i$ and the action from the behavior policy.
>
> Note that both terms are bounded, and the difference between the second and third terms is no less than $-\frac{\tau}{1-\tau}|A_i|(a_{\max}-a_{\min})^2$, where $\tau \in (0,1)$ is the regularization coefficient, $|A_i|$ denotes the dimension of agent $i$'s action space, and $a_{\max}$ and $a_{\min}$ represent the maximum and minimum actions. Since the term can be controlled, we find that OMAR gives a safe policy improvement guarantee over $\hat{\pi}_{\beta_i}$.
>
> *Q2: The experiments largely focus on cases where there are a few agents. Authors should include a curve showing how the method compares as the number of agents increases to a large number.*
>
> - Figure 2 in the paper demonstrates the performance improvement percentage of OMAR over MA-CQL with an increasing number of agents, i.e., $\frac{score_{OMAR} - score_{MA-CQL}}{score_{MA-CQL}} \times 100\%$​​​. The result shows that OMAR outperforms MA-CQL by a larger margin in multi-agent tasks compared to the single-agent setting.
>
> - For the larger-scale tasks, we also include an additional set of experiments in the challenging StarCraft II micromanagement benchmark [1] in Appendix B.3 (based on the gumbel-softmax estimator to be applicable in tasks with discrete action spaces) with a range of $5$ to $66$ agents including 2s3z, 3s5z, 1c3s5z, and 2c_vs_64zg. Figure 7 in Appendix B.3 demonstrates the results of OMAR with an increasing number of agents and difficulties. As shown, OMAR significantly outperforms MA-CQL, and further confirms that our method can scale to large-scale tasks.
>
> [1] Samvelyan, Mikayel, et al. "The starcraft multi-agent challenge." arXiv preprint arXiv:1902.04043 (2019).

---

> ### Author Response · Authors · 2021-11-29
> **Follow up on Rebuttal**
>
> Dear Reviewer pt8q,
>
> Thank you again for your review! Could you please kindly let us know whether the responses and the updates in the paper (marked in blue) address your concerns? We are happy to provide further clarification if you have any additional concerns.
>
> Thanks and looking forward to your reply!

---

### Official Review · Reviewer_cYK5 · 2021-11-02

**Correctness:** 3
**Technical Novelty And Significance:** 3
**Empirical Novelty And Significance:** 3
**Recommendation:** 5
**Confidence:** 4

**Main Review:**

## strengths

The local optimum problem is successfully solved using zero-order optimization for offline training in the multi-agent scenario for the first time.

The paper is clear and easy to understand, and the approach is simple and effective.

## weaknesses


As mentioned in the paper, offline learning on multi-agent tasks is very common but poorly studied. Although the approach proposed in this paper achieves significant performance, the following issues are not explained or investigated very clearly.

1. Regarding the motivation, although MAPPO is successful, it is undeniable that derictly applying the single-agent algorithm to multi-agent tasks may probabily perform not well (e.g. indepdent Q learning). So it is not surprising that MA-CQL performance is poor. The authors claim that this is due to the local optimum, but there is not sufficient experimental arguments to illustrate that. The simple illustration given in Figure 1.d may not appropriate to represent the relationship between the local optimum and the global optimum, because a higher Q value may be caused by overestimation.

2. Regarding the method, zero-order optimization can be applied not just under multi-agent tasks. So I think a key ablation experiment to illustrate the contribution of zero-order optimization to offline reinforcement learning is performing zero-order optimization based on single-agent CQL.

3. There are some details about the methodology that are worth discussing.
3.1 Why does zero-order optimization not consider the minimum of multiple Q's?
3.2 The authors argue that solely regularizing the critic is insufficient for multiple agents to learn good policies for coordination in the offline setting, but in fact, the authors do not analyze coorparation explitily. So is this really the key issue of CQL?

4. There are also some details about the experiment that are worth discussing.
4.1 Why the performance variance in Figure 3 is so large.
4.2 Why is the learning rate set so large (0.01) in the training process? An explanation is needed, since the learning rate is usually set to 3e-4 in CQL or SAC.
4.3 In Table 5, why is the performance worse under the CTDE paradigm? This is somehow contradictory to the conclusion of MADDPG, so a more sufficient explanation is needed.

Overall, this paper addresses important question, but is not well researched. So, I consider the paper to be at the level of borderline for now. I would be happy to raise the score if the authors could more clearly verify the motivation through experiments, and discuss the advantages that zero-order optimization can bring under single-agent offline reinforcement learning tasks. I look forward to the follow-up of this work.


**Summary Of The Paper:**

This paper proposes a simple yet effective multi-agent offline reinforcement learning algorithm, called Offline Multi-Agent RL with Actor Rectification, which combines CQL (first-order optimization) and evolutionary algorithms (zero-order optimization) to enhance the applicability of the CQL algorithm on multi-agent tasks.

**Summary Of The Review:**

Overall, this paper addresses important question, but is not well researched. So, I consider the paper to be at the level of borderline for now, and I recommend marginally below the acceptance threshold (5).

---

> ### Author Response · Authors · 2021-11-21
> **Author Response (Part 1 of 2)**
>
> We thank the reviewer for the thoughtful review and detailed evaluation of our paper, and for noting that we "addressed an important question", "proposed a simple yet effective approach with significant performance", and "the paper is clear and easy to understand".
>
> *Q1: The simple illustration given in Figure 1.d may not be appropriate to represent the relationship between the local optimum and the global optimum, because a higher Q value may be caused by overestimation.*
>
> Conservative Q-learning, which is the base algorithm of our method, learns a Q-function that lowers-bounds the true Q-function and largely reduces overestimation (Table 4 in [1]). In addition, the policy in OMAR is designed to maximize the given estimated Q-value in CQL or other related Q-learning based algorithms. Therefore, even if Q-value is overestimated, this should be considered as a failure of Q-value estimation, but does not justify the convergence of the policy to the local optimum of the Q-value.
>
> *Q2: Regarding the method, zero-order optimization can be applied not just under multi-agent tasks. So I think a key ablation experiment to illustrate the contribution of zero-order optimization to offline reinforcement learning is performing zero-order optimization based on single-agent CQL.*
>
> Besides the single-agent setting of the Spread task we have shown in Figure 2, we also include an additional experiment in Appendix B.6 to evaluate the effectiveness of our method in single-agent tasks by comparing it with CQL in the Maze2D domain from the D4RL benchmark [2].
>
> Table 2 below shows the results in an increasing order of complexity of the maze, where OMAR also outperforms CQL. More details for the task can be found in Appendix B.6.
>
> | | maze2d-umaze | maze2d-medium | maze2d-large |
> | :------: | :------: | :------: | :------: |
> | CQL | $109.8\pm23.9$ | $106.4\pm11.0$ | $94.6\pm44.6$ |
> | OMAR | $\textbf{124.7}\pm7.6$ | $\textbf{125.7}\pm12.3$ | $\textbf{157.7}\pm12.3$ |
>
> Table 2. Averaged normalized score of OMAR and CQL in the single-agent Maze2D domain.
>
> We also include a discussion about the effectiveness of OMAR in single-agent and multi-agent tasks in Appendix B.4, where OMAR is effective in both settings, but the performance gain is more significant in the challenging offline multi-agent setting.
>
> *Q3.1: Why does zero-order optimization not consider the minimum of multiple Q’s?*
>
> We think the reviewer means the consideration of an ensemble of Q-functions, and taking their minimum in the zeroth-order optimizer part. Please kindly correct us if we misunderstand your point. Considering the minimum of multiple Q’s in this way is similar to Maxmin Q-learning [3], which aims to reduce the overestimation bias. We believe the proposed method can be complementary yet tangential to our method. In practice, we find that overestimation is not the outstanding issue for our algorithm since we are based on CQL that learns a conservative value function and lower-bounds the expected value. So we did not add other techniques in the zeroth-order optimizer.
>
> *Q3.2: The authors argue that solely regularizing the critic is insufficient for multiple agents to learn good policies for coordination in the offline setting, but in fact, the authors do not analyze cooperation explicitly. So is this really the key issue?*
>
> We include an additional experiment in Appendix B.7 to clarify this point. Figure 10 in Appendix B.7 shows the result of the performance improvement percentage of a non-cooperative version of the Spread task that involves multiple agents with the same setup as the original task except for the independent reward functions (i.e., arriving at its own target landmark). As shown in Figure 10, the performance of CQL does not degrade with an increasing number of agents in this setting where cooperation is not needed.  However, its performance degrades dramatically in the cooperative Spread task as there are more agents in Figure 1(c). This observation confirms that the issue is due to the failure of coordination.

---

> > ### Author Response · Authors · 2021-11-21
> > **Author Response (Part 2 of 2)**
> >
> > *Q4.1: Why the performance variance in Figure 3 is so large.*
> >
> > We limit the range of the y-axis to make it centered around the dots originally intended for a clearer view, since Figure 3 summarizes the results in different tasks. The variance is not around the same scale as in other experimental results. We have updated the plot in Figure 3 in the paper to avoid confusion.
> >
> > *Q4.2: Why is the learning rate set so large (0.01) in the training process? An explanation is needed, since the learning rate is usually set to 3e-4 in CQL or SAC.*
> >
> > We use the default fine-tuned learning rate as in MADDPG [4] in multi-agent particle environments, which is comparatively larger than the learning rate in SAC or CQL in MuJoCo tasks. We have also tried smaller learning rates but they are underperforming as shown in Table 3 below, so we use the default fine-tuned hyperparameters as in MADDPG for the particle tasks.
> >
> > |Learning rate | $1e-4$ | $3e-4$ | $5e-4$ | $1e-3$ | $3e-3$ | $5e-3$ | $1e-2$ |
> > | :------: | :------: | :------: | :------: | :------: | :------: | :------: | :------: |
> > |Performance | $140.2\pm21.2$ | $148.7\pm15.2$ | $152.3\pm17.1$ | $164.0\pm14.5$ | $230.4\pm31.1$ | $256.2\pm34.2$ | $\textbf{267.9}\pm19.0$ |
> >
> > Table 3. Performance of MA-CQL with smaller learning rate.
> >
> > *Q5. In Table 5, why is the performance worse under the CTDE paradigm? This is somehow contradictory to the conclusion of MADDPG, so a more sufficient explanation is needed.*
> >
> > We attribute the lower performance in Table 5 (based on centralized value functions) compared to Table 3 (based on decentralized value functions) due to the base algorithm, where Table 9 in Appendix B.8 shows the performance comparison of offline independent TD3 and offline multi-agent TD3. As shown, utilizing centralized critics underperforms decentralized critics in the offline setting. There has also been recent research [5, 6] showing the benefits of decentralized value functions compared to a centralized one, which leads to a more robust performance. We attribute the performance loss of CTDE in the offline setting due to a more complex and higher-dimensional value function conditioning on all agent's actions and the global state that is harder to learn well without exploration.
> >
> > [1] Kumar, Aviral, et al. "Conservative q-learning for offline reinforcement learning." arXiv preprint arXiv:2006.04779 (2020).
> >
> > [2] Fu, Justin, et al. "D4rl: Datasets for deep data-driven reinforcement learning." arXiv preprint arXiv:2004.07219 (2020).
> >
> > [3] Lan, Qingfeng, et al. "Maxmin q-learning: Controlling the estimation bias of q-learning." arXiv preprint arXiv:2002.06487 (2020).
> >
> > [4] Lowe, Ryan, et al. "Multi-agent actor-critic for mixed cooperative-competitive environments." arXiv preprint arXiv:1706.02275 (2017).
> >
> > [5] de Witt, Christian Schroeder, et al. "Is Independent Learning All You Need in the StarCraft ulti-Agent Challenge?." arXiv preprint arXiv:2011.09533 (2020).
> >
> > [6] Lyu, Xueguang, et al. "Contrasting centralized and decentralized critics in multi-agent reinforcement learning." arXiv preprint arXiv:2102.04402 (2021).

---

> > > ### Comment · Reviewer_cYK5 · 2021-11-30
> > > **I keep my score**
> > >
> > > Thanks for providing the clarifications. Based on the overall discussion, I believe that the paper could be considerably improved, so I will keep my score as it is.

---

> ### Author Response · Authors · 2021-11-29
> **Follow up on Rebuttal**
>
> Dear Reviewer cYK5,
>
> Thank you again for your review! Could you please kindly let us know whether the responses and the updates in the paper (marked in blue) address your concerns? We are happy to provide further clarification if you have any additional concerns.
>
> Thanks and looking forward to your reply!

---

### Official Review · Reviewer_6gYu · 2021-11-02

**Correctness:** 4
**Technical Novelty And Significance:** 3
**Empirical Novelty And Significance:** 3
**Recommendation:** 6
**Confidence:** 3

**Main Review:**

The authors proposed a simple but seemingly effective method for improving MARL with offline datasets. The paper is overall well written and easy to follow. I think the paper is looking at an important problem and contributing to the development of offline MARL, however, I have a few questions:

1. OMAR uses a regularizer to encourage the actor policy to take action with a high Q-value. However, if the local Q-value gradient direction is opposite compared to the selected sampled action, would the actor policy get stuck in some suboptimal actions (e.g. in the middle of the two high Q-value points).
2. Regarding using the Gaussian function sampling, I wonder if its performance would be affected by the dataset. Since CQL tries to reduce the Q-values of unknown actions, depending on the dataset, if the value function landscape has multiple picks (e.g. mix Gaussian), then the sampling method may result in suboptimal action. Can this issue be handled by this method?
3. Does this value function saddle issue happens only in the offline setting? Can you provide some intuition regarding why the issue is more severe in an offline setting?
4. It would be nice to also include the score of the online learning for the expert dataset.
5. How does the size of the training dataset affects the benefit of this method. Does the low optimal issue go away when you have more data points?


**Summary Of The Paper:**

This work looks at the problem of training multi-agent reinforcement learning with continuous action space in an offline setting. The authors identified a saddle point issue of the value function landscape in the existing offline MARL/RL methods, which causes the actor policy to be stuck in a bad local optimum. The proposed method samples and evaluates different actions based on a Gaussian function, and adds a regularizer to the actor loss to encourage the actor policy to take the action with a high Q-value.


**Summary Of The Review:**

The authors identified a concrete problem in offline MARL training and proposed a method that is addressing the issue. The evaluation shows that overall the proposed method outperforms the existing baselines.

---

> ### Author Response · Authors · 2021-11-21
> **Author Response (Part 1 of 2)**
>
> We thank the reviewer for the helpful review and positive assessment of our work, and for noting that we studied "an important problem" and proposed "a simple yet effective method" in a "well-written and easy to follow way".
>
> *Q1: If the local Q-value gradient direction is opposite compared to the selected sampled action, would the actor policy get stuck in some suboptimal actions (e.g., the middle of the two high Q-value points)?*
>
> Indeed, if the gradient direction is opposite, there might be such issues. However, in practice we find the optimization is usually successful, resulting in significant performance gain in different types of datasets and different tasks. We perform optimization multiple times. For each time, we sample the actions so that the chance that the direction between the two gradients is always opposite is low. We are the first to utilize zeroth-order optimization in offline MARL settings, and it is an interesting direction for future work to follow your suggestion and investigate better paradigms such as iterative optimization between the two terms or adaptive/learnable weight for the objective.
>
> *Q2: If the value function landscape has multiple peaks (e.g., mix Gaussian), then the sampling method may result in suboptimal action. Can this issue be handled by the Gaussian function sampling method? Is the performance affected by the dataset?*
>
> Thanks for your insightful comment. We guarantee that the action from the zeroth-order optimizer is better than the action from CQL. In addition, the mean of the Gaussian distribution is iteratively updated by Eq. (3), which is a softer update that leverages more samples from different peaks compared to the cross-entropy method (CEM), and our sampling mechanism also outperforms CEM as shown in Table 4 in the paper. It is an interesting direction for utilizing a more general class of distributions to represent more complex and multimodal behaviors for future work (e.g., energy-based policies).
>
> *Q3: Does this value saddle issue happen only in the offline setting? Can you provide some intuition regarding why the issue is more severe in an offline setting?*
>
> This also presents a challenge in the online setting, but is more severe in the offline case. We conduct an additional experiment in the online setting where our method is still effective in Appendix B.4. We also include a more detailed discussion to illustrate this point.
>
> - (Additional experiments) We include an additional experiment which evaluates the effectiveness of our proposed learning objective in the online setting. The experimental setting is the same as in Figure 2 in the paper except that we consider the online case based on the multi-agent TD3 algorithm.
> Table 1 below summarizes the performance improvement percentage of our method over the baseline with a varying number of agents in the online setting. As shown, our method is still effective, but with a comparatively smaller improvement margin compared to Figure 2.
>
> | $N=1$ | $N=2$ | $N=3$ | $N=4$ | $N=5$ |
> | :------: | :------: | :------: | :------: | :------: |
> | $2.7\%$ | $17.5\%$ | $36.9\%$ | $24.8\%$ | $20.3\%$ |
>
> Table 1. Performance improvement percentage of our method in the online setting over multi-agent TD3 with varying number of agents in the Spread environment.
>
> - (Intuition) While the problem also happens in the online case, we observe that this is much more challenging in the offline setting, since agents cannot explore and interact with the environment. Intuitively, in the online setting,  if the actor has not well exploited the global information in the value function, it can still interact with the environment and collect better experiences for improving the estimation of the value function and providing a better guidance for the policy. However, no exploration and interaction with the environment for new data collection are allowed in the offline setting. Thus, it is much harder for an agent to escape from a bad local optimum and difficult for the actor to best leverage the global information in the critic. In addition, this presents an even more challenging problem in the offline multi-agent setting because multiple agents result in an exponentially-sized joint action space as well as the nature of the setting that requires a coordinated joint policy. A more detailed analysis can be found in Appendix B.4.
>
> *Q4: It would be nice to include the score of the online learning for the expert dataset.*
>
> Thanks for the suggestion, we have included the expert and random scores in Appendix B.1.

---

> > ### Author Response · Authors · 2021-11-21
> > **Author Response (Part 2 of 2)**
> >
> > *Q5: How does the size of the training dataset affect the benefit of this method? Does the low optimal issue go away when you have more data points?*
> >
> > We include an additional ablation study in Appendix B.5 to study the effect of the size of the dataset following the experimental protocol in [1]. Specifically, we first generate a full replay dataset by recording all samples in the replay buffer encountered during the training course for $1$ million steps in the cooperative navigation task. Then, we randomly sample $N\\%$ experiences from the full replay dataset and obtain several smaller datasets with the same data distribution.
> >
> > Figure 9 in Appendix B.5 shows that the performance of MA-CQL increases given more data points for $N\in\\{1,10,20\\}$. However, its performance does not further increase given an even larger amount of data, which performs much worse than the fully-trained online agents and fails to recover their performance. On the contrary, OMAR always outperforms MA-CQL by a large margin when $N>1\\%$, whose performance is much closer to the fully-trained online agents given more data points. Therefore, the optimality issue still persists (e.g., it can take a very long time to escape from them if the objective contains very flat regions [2]). In addition, the zeroth-order optimizer part in OMAR can better guide the actor given a larger amount of data points with a more accurate value function.
> >
> > [1] Agarwal, Rishabh, Dale Schuurmans, and Mohammad Norouzi. "An optimistic perspective on offline reinforcement learning." International Conference on Machine Learning. PMLR, 2020.
> >
> > [2] Ahmed, Zafarali, et al. "Understanding the impact of entropy on policy optimization." International Conference on Machine Learning. PMLR, 2019.

---

> ### Author Response · Authors · 2021-11-29
> **Follow up on Rebuttal**
>
> Dear Reviewer 6gYu,
>
> Thank you again for your review! Could you please kindly let us know whether the responses and the updates in the paper (marked in blue) address your concerns? We are happy to provide further clarification if you have any additional concerns.
>
> Thanks and looking forward to your reply!

---

### Official Review · Reviewer_qETT · 2021-11-03

**Correctness:** 4
**Technical Novelty And Significance:** 2
**Empirical Novelty And Significance:** 2
**Recommendation:** 6
**Confidence:** 3

**Main Review:**

I thought that the paper was pretty well written and did a pretty good job of motivating the fact multi-agent optimization is more likely to fall into bad local minima than single agent optimization. However, I am only so convinced by empirical examples and think the paper can be improved by motivating this fact more based on the theory of MARL i.e. the non-stationary underlying optimization process. I did appreciate, however, the theoretical contribution of Theorem 1 demonstrating that the approach leads to sound updates. Zeroth order optimization has been considered in the RL literature, so its application is not hugely novel in this context. Moreover, I can't shake the feeling that zeroth order optimization is somewhat indirectly related to the multi-agent/offline RL problems specifically.

The experiments were relatively small scale, but these things are harder in the multi-agent literature and the complexity considered is fairly respectable in comparison to related prior work. I found the experiments provided to be relatively thorough, addressing a lot of key questions I had about the approach. One question I had though was if the bolds in Table 3 are merely higher numbers or statistically significant.



**Summary Of The Paper:**

This paper considers the offline multi-agent RL setting, first demonstrating that optimization is more likely to find bad local optima than in the single agent case. To deal with this problem, the authors propose to add zeroth order optimization to multi-agent training and provide a theorem guaranteeing that this approach leads to safe improvements. The authors conduct extensive experiments and ablations in comparison to relevant baselines on the multi-agent particle environments to demonstrate the efficacy of their approach as a function of the type of data used for offline training. The authors also provide experiments on a slightly larger scale tackling the multi-agent half-cheetah environment.

**Summary Of The Review:**

I generally like this paper and am in favor of acceptance. I think the paper presents a pretty good narrative and addresses the noted optimization issues with a somewhat novel and theoretically grounded approach. The experiments are pretty comprehensive although admittedly of very limited scale. What stops me from giving a higher score is mostly the somewhat disconnected nature of the proposed contribution to the theory of MARL, which is at this point mostly based on intuitive/empirical findings.

---

> ### Author Response · Authors · 2021-11-21
> **Author Response**
>
> We thank the reviewer for the useful feedback and positive assessment of our work, and for noting that "we did a pretty good job in the motivation", "conducted extensive experiments", and "the paper was well-written".
>
> *Q1: The paper can be improved by motivating the fact that “multi-agent optimization is more likely to fall into bad local optima than single-agent optimization” based on the theory of MARL, i.e., the non-stationary underlying optimization process.*
>
> We thank the reviewer for the insightful comments. Indeed, multi-agent optimization has been shown to be theoretically much more challenging---it is well-known that finding approximate Nash equilibrium of general two-player games is considered to be computationally hard (or more precisely, proved to be PPAD-hard) [1]. This suggests that it is even difficult to make sure each agent is playing optimally while other agents are fixed. Moreover, the environment for each individual player is constantly changing because the other players are being updated, which exacerbates the problem. We will add these discussions to the revision. On the other hand, we also noted that a rigorous analysis of the training dynamics is extremely challenging because all the optimization objectives are non-convex.
>
> *Q2: Experiments were relatively small scale.*
>
> We include an additional set of experiments in the challenging StarCraft II micromanagement benchmark [2] (based on the gumbel-softmax estimator to be applicable in tasks with discrete action spaces) including 2s3z, 3s5z, 1c3s5z, and 2c_vs_64zg with an increasing number of agents (from $5$ to $66$) and difficulties in Appendix B.3. Figure 7 in Appendix B.3 shows that OMAR significantly outperforms MA-CQL on all tested maps, with an average performance gain of $76.7\\%$. The result further confirms that our method can scale to more challenging and large-scale tasks.
>
> *Q3: If the bolds in Table 3 are merely higher numbers or statistically significant?*
>
> The bolds are statistically significant, and learning curves can be found in Figure 6 in Appendix B.2 where the shaded region corresponds to the standard deviation.
>
> *Q4: I cannot shake the feeling that zeroth order optimization is somewhat indirectly related to the multi-agent/offline RL problems specifically.*
>
> We include an additional experiment in Appendix B.4 to investigate the effectiveness of our method in the online/single-agent settings.
>
> (Result) As shown in Figure 8 in Appendix B.4, our method is generally applicable in reinforcement learning including online RL, offline RL, multi-agent RL, and offline multi-agent RL settings, with the most significant performance gain in the particularly challenging offline multi-agent RL case.
>
> (Discussion) In the offline setting, the performance improvement is much more significant than the online setting, because the agents cannot explore and interact with the environment. Intuitively, in the online setting,  if the actor has not well exploited the global information in the value function, it can still interact with the environment to collect better experiences for improving the estimation of the value function and provides a better guidance for the policy. However, no exploration and interaction with the environment for new data collection are allowed in the offline setting. Thus, it is much harder for an agent to escape from a bad local optimum and difficult for the actor to best leverage the global information in the critic.
> This presents an even more challenging problem in MARL because multiple agents result in an exponentially-sized joint action space as well as the nature of the setting that requires a coordinated joint policy. As expected, we also find that the performance gain is more significant in the offline multi-agent domain, which requires each of the agents to learn a good policy for a successful joint policy for coordination. Otherwise, it can lead to an uncoordinated global failure.
>
> [1] Daskalakis, Constantinos. "On the complexity of approximating a Nash equilibrium." ACM Transactions on Algorithms (TALG) 9.3 (2013): 1-35.
>
> [2] Samvelyan, Mikayel, et al. "The starcraft multi-agent challenge." arXiv preprint arXiv:1902.04043 (2019).

---

> ### Author Response · Authors · 2021-11-29
> **Follow up on Rebuttal**
>
> Dear Reviewer qETT,
>
> Thank you again for your review! Could you please kindly let us know whether the responses and the updates in the paper (marked in blue) address your concerns? We are happy to provide further clarification if you have any additional concerns.
>
> Thanks and looking forward to your reply!

---

### Comment · Area_Chair_2VJD · 2021-11-20
**Please read other reviews**

Dear reviewers,

The authors have not provided any responses, but could you please read the reviews by others to see how your evaluation changes.  Thank you.

---

### Author Response · Authors · 2021-11-21
**Summary of Changes**

We thank all reviewers for their useful feedback that has helped in improving our paper. We greatly appreciate the thoughtful reviews!

We have updated the paper to address reviewers' comments, where the revisions are marked in blue. In summary, the major updates we have made to the paper include:
- Appendix B.3: We add an experiment of OMAR in the challenging StarCraft II micromanagement benchmark to demonstrate its effectiveness in larger-scale tasks.
- Appendix B.4: We add an experiment of OMAR in online RL, online multi-agent RL, offline RL, and offline multi-agent RL settings to study its effectiveness in different settings. Results show that our method is generally applicable in different settings, with a much more significant performance gain in the particularly challenging offline multi-agent setting. We also include a detailed discussion about why the issue is more severe in the offline multi-agent case.
- Appendix B.5: We add an ablation study to investigate the effect of the size of the dataset.
- Appendix B.6: We add an experiment for evaluating OMAR in single-agent Maze2D tasks from the D4RL benchmark.
- Appendix B.7: We add an experiment of MA-CQL in a non-cooperative setting of the Spread task to emphasize the effect of coordinated learning behavior.
- Appendix B.8: We include the comparison result of offline independent TD3 and offline multi-agent TD3 and a more detailed discussion about their performance.

We would very much appreciate it if the reviewers can check our responses and the updates in the paper and hope that they address your concerns. We are happy to provide further clarification if you have any additional concerns.

---

### Decision · Program_Chairs · 2022-01-20

**Decision:**

Reject

**Comment:**

This paper makes a key observation that the gradient-based method gets more likely to suffer from poor local optima in multi-agent reinforcement learning (MARL) with more agents particularly in the offline setting.  The paper proposes the use of zeroth order optimization method to avoid local optima.  Specifically, it samples multiple actions and regularize the policy to get closer to the optimal action among those.  The use of such zeroth order method to avoid poor local optima is not particularly new, although finding its effective in MARL and the empirical support are valuable.  The main discussion point was the insufficiency of experimental support, and the additional experiments during the discussion have addressed the original concerns of the reviewers to some extent.  Overall, given the limited novelty and inefficiency of support (either theoretical or empirical), the paper is slightly below the borderline.